# Boosting Norwegian Automatic Speech Recognition

**Javier de la Rosa**
versae@nb.no

**Rolv-Arild Braaten**
rolv.braaten@nb.no

**Per Egil Kummervold**
per.kummervold@nb.no

**Freddy Wetjen**
freddy.wetjen@nb.no

**Svein Arne Brygfjeld**
svein.brygfjeld@nb.no

National Library of Norway, Norway

## Abstract

In this paper, we present several baselines for automatic speech recognition (ASR) models for the two official written languages in Norway: Bokmål and Nynorsk. We compare the performance of models of varying sizes and pre-training approaches on multiple Norwegian speech datasets. Additionally, we measure the performance of these models against previous state-of-the-art ASR models, as well as on out-of-domain datasets. We improve the state of the art on the Norwegian Parliamentary Speech Corpus (NPSC) from a word error rate (WER) of 17.10% to 7.60%, with models achieving 5.81% for Bokmål and 11.54% for Nynorsk. We also discuss the challenges and potential solutions for further improving ASR models for Norwegian.

## 1 Introduction

Automatic Speech Recognition (ASR) is the task of converting speech into text. ASR systems are used in a wide range of applications, such as voice assistants, transcription services, and speech-to-text translation. It is also increasingly becoming a tool for research in spoken language as the accuracy of the more recent neural-based models is approaching that of humans for certain metrics. In a study by Amodei et al. (2016), the authors estimated that the word error rate (WER) in human-produced transcriptions on the LibriSpeech benchmark (Panayotov et al., 2015) is roughly 5.83%, while their end-to-end ASR model, DeepSpeech 2, achieved a WER of 5.33% on a clean test set, although it was outperformed by humans on noisy data. Since the introduction of DeepSpeech 2, the field of ASR has progressed even further, with the current leaderboard of the benchmark containing

over ten models with a WER below 2%. Despite the high accuracy in resource-rich languages, ASR models are currently unavailable for the vast majority of the world's languages due to the lack of gold annotated data to train such models. Recent advances in unsupervised learning of acoustic models have decreased the need for transcribed speech.

In this paper, we focus on developing and evaluating a new set of baselines ASR models for Norwegian based on the wav2vec 2.0 architecture (Baevski et al., 2020). We make use of existing pre-trained models and combine them with other language resources for the Norwegian languages to further improve the accuracy of the resulting ASR systems. Our models seem to perform notably better than previous work on newly established datasets.

## 2 Norwegian ASR

The Norwegian language has many spoken dialects, which differ lexically, grammatically, and phonologically. Additionally, there are two official written standards of Norwegian, Bokmål and Nynorsk, which have somewhat different inflection, vocabulary, and spelling. Consequently, high-quality datasets for acoustic modeling of Norwegian require speech data in different dialects and should ideally include transcriptions in both written standards.

Early work on Norwegian speech recognition was mostly focused on very limited vocabularies and numbers, tailored for telephone applications and menu navigation (Svendsen et al., 1989; Paliwal, 1992; Ljøen et al., 1994; Kvale, 1996). Compound words are more frequent in Norwegian than English, but using traditional pronunciation dictionaries seemed sufficient in controlled lexicons. In Norwegian, natural numbers between 20 and 99 can be pronounced differently (e.g. "twenty-four" and "four-and-twenty"), which poses a chal-

lenge for natural number recognition. By the year 2000, and under the umbrella of a few EU-funded projects, research focused mostly on overcoming these limitations and extending the use cases to dates, times, nouns, and the spelling out of words, which yielded several important datasets (e.g., SpeechDat, SpeechDat-II, TABU.0) and technical improvements over a short period of time (Amdal and Ljøen, 1995; Hoge et al., 1997; Kvale and Amdal, 1997; Johansen et al., 1997; Amdal et al., 1999; Martens, 2000). Most approaches were based on hidden Markov models and some relied on Mel Frequency Cepstral Coefficients (MFCC), commonly by using the Hidden Markov Model Toolkit (HTK) (Young and Young, 1993).

However, these approaches were not designed for open-ended recognition and often struggled with out-of-vocabulary words or real conversations. It was not until the introduction of newer datasets in the last decade that systems with reasonable performance started to appear.

## 2.1 NST

The Nordisk Språkteknologi (NST) dataset is a multi-lingual speech recognition dataset with speech in Swedish, Danish and Norwegian Bokmål, and their corresponding transcriptions. Developed by the now extinct technology company Nordisk Språkteknologi in the late 90s and beginning of the 2000s, the data was manually compiled and mostly validated. It contains telephone conversations, office conversations, read aloud passages, word spellings, and even hesitations. The speaker metadata includes age, gender, region of birth, and regional dialect. The audio quality is generally high, and most recordings have two channels recorded with separate microphones, one placed close to the speaker and one across the room. The dataset comes with training and testing sets. For Norwegian, the training set contains 411.5 hours of speech, while the test contains 115.3 hours. The amount of speech in hours per the regional dialect of the speakers represented in the NST dataset is reported in Table 9 of Appendix C. However, due to its nature as a manuscript-read dataset, the dataset has some limitations, as it only contains planned speech and does not include or contains limited degree of dialectal phenomena which deviate from the Bokmål norm.

## 2.2 NPSC

In Solberg and Ortiz (2022), the authors present the Norwegian Parliamentary Speech Corpus (NPSC, The National Library of Norway, 2021), an open dataset intended for acoustic modeling of Norwegian unscripted speech. The dataset is developed and distributed by the Language Bank at the National Library of Norway, and consists of approximately 100 hours of recordings of meetings at Stortinget, the Norwegian parliament, in 2017 and 2018. Orthographic transcriptions in Norwegian Bokmål and Norwegian Nynorsk were made. The dataset is public domain and can be used with no restrictions. The dataset is split in training, validation, and test sets (see Table 1).

Solberg and Ortiz trained and tested an ASR system and the results showed that the use of the NPSC dataset improved the recognition performance when compared to the use of only manuscript-read datasets. The authors argue that the NPSC dataset is necessary to fill the gap in the lack of available speech data for Norwegian ASR.

## 2.3 FLEURS

A very recent addition to the small pool of open datasets suitable for training transformer-based models for ASR comes in the form of a multilingual speech benchmark. The Few-shot Learning Evaluation of Universal Representations of Speech (FLEURS) benchmark (Conneau et al., 2022) is a parallel speech dataset in 102 languages built on top of the FLoRes-101 benchmark for machine translation. FLEURS contains approximately 12 hours of speech per language and can be used for various speech tasks such as automatic speech recognition, speech language identification, translation, and retrieval. The goal of FLEURS is to enable speech technology in more languages and drive research in low-resource speech understanding. The dataset is unique in its coverage of over 100 languages and its suitability for various speech tasks. In their paper, the authors provide baseline results for the different tasks using multilingual pre-trained models, but do not report on single monolingual ones. The almost 11 hours of Norwegian (see Table 2) included in this dataset adhere to Bokmål and represent out of domain speech qualitatively closer to NST than to NPSC.

| Language | Train | | Validation | | Test | |
|---|---|---|---|---|---|---|
| | **Hours** | **Samples** | **Hours** | **Samples** | **Hours** | **Samples** |
| Norwegian Bokmål | 88.62 | 44,746 | 11.70 | 5,973 | 11.15 | 5,527 |
| Norwegian Nynorsk | 12.96 | 6,586 | 1.61 | 871 | 1.33 | 828 |
| **Total** | **101.58** | **51,332** | **13.31** | **6,844** | **12.48** | **6,355** |

Table 1: Distribution of number of hours and samples for each of the Norwegian written languages in the NPSC dataset.

| Train | | Validation | | Test | |
|---|---|---|---|---|---|
| **Hours** | **Samples** | **Hours** | **Samples** | **Hours** | **Samples** |
| 10.91 | 3,167 | 0.58 | 163 | 1.25 | 357 |

Table 2: Distribution of number of hours and samples for each of the splits in Norwegian subset of the FLEURS dataset.

## 3   Norwegian wav2vec 2.0

Introduced by Baevski et al. (2020), wav2vec 2.0 is a state-of-the-art self-supervised audio representation learning architecture designed to extract high-quality feature representations from raw audio signals. After pre-training the acoustic model, wav2vec 2.0 models can be used for a wide range of tasks using a regular fine-tuning mechanism. For ASR, these fine-tuned models can be plugged to rather simple n-gram language models that leverage the connectionist temporal classification (CTC) classification loss to further improve recognition.

Wav2vec 2.0 improves upon the original wav2vec architecture by Schneider et al. (2019) in several key ways. First, it uses a transformer-based neural network to predict the audio signal in a context window surrounding a masked center frame. This enables the model to capture long-range dependencies in the audio signal, leading to more accurate feature representations. Second, the model performs multiple prediction tasks simultaneously, including predicting the center frame, predicting the entire context window, and predicting future audio signals. The CTC loss is used to compute the prediction error between the predicted and actual center frame. This multi-task learning approach improves the representational power of the model. Finally, wav2vec 2.0 has a larger number of parameters and a larger training data size, which leads to improved performance on various audio representation learning benchmarks.

In early 2022, we released a series of wav2vec 2.0 models of different sizes. Available for

Bokmål in 300 million[1] and 1 billion[2] sizes and for Nynorsk only in 300 million parameters[3], these models were fine-tuned on the NPSC dataset. The 1 billion parameter models were based on the multilingual XLS-R models, and the 300 million parameters models on the Swedish VoxRex model. XLS-R models (Babu et al., 2021) are trained on more than 436,000 hours of publicly available speech recordings. The data used to train the XLS-R models came from a variety of sources, including parliamentary proceedings and audio books, and covered 128 different languages. VoxRex, developed by Malmsten et al. (2022) at National Library of Sweden (KB), is a Swedish acoustic wav2vec 2.0 model trained on the P4-10k corpus which contains 10,000 hours of Swedish local public service radio as well as 1,500 hours of audio books and other speech from KB's collections. The choice of a Swedish acoustic model to fine-tune Norwegian ASR instead of using the same size XLS-R model was motivated by the fact that both languages belong to the North Germanic language family, which all originated from Old Norse, and share many spoken and written features.

## 4   Methods

In this work, we evaluate these models, referred to as NPSC-Bokmål and NPSC-Nynorsk, and fine-tune new XLS-R 1 billion (1B) parameters and VoxRex 300 million (300M) parameters models

---

[1]https://huggingface.co/NbAiLab/nb-wav2vec2-300m-bokmaal
[2]https://huggingface.co/NbAiLab/nb-wav2vec2-1b-bokmaal
[3]https://huggingface.co/NbAiLab/nb-wav2vec2-300m-nynorsk

using the same hyperparameters[4]. We train the models on NPSC and ablate on different data supplementing strategies derived from the NST dataset.

The NST dataset was modernized and re-organized by the National Library of Norway, and is now available in a reader-friendly format (Nordisk Språkteknologi, 2020). We omitted the second channel of audio recorded with a distant microphone due to no noticeable differences between the audio recorded with the close microphone. The dataset is representative of the major regions and the language variety spoken in that region, although the representation of the dialectal varieties of the Scandinavian languages in the dataset is debatable (see Appendix C, Table 9). All combinations of NPSC and NST training sets were lowercased, and had removed non-letter characters and accents from characters (aside from the Norwegian 'æøå'). Any samples with an audio clip under half a second are removed. Transcripts containing digits are also removed, as we expect any numbers to be spelled out. NST data containing words spelled out letter by letter were removed, and instructions to stay silent or dictation commands (e.g., comma, period) were replaced with empty strings. For the hesitations in NPSC and NST, most of the runs replace them using triple letters, e.g. `<ee>` becomes `eee`. These models also use the Bokmål translation of the Nynorsk data in NPSC. The resulting models from the different experiments are listed below:

- NST model. Fine-tuned on the NST dataset as described, with no exta modificatons nor additions.

- NST-NPSC model. These models are fine-tuned using the Bokmål and Nynorsk subsets of NPSC plus the NST dataset as described.

- NST-NPSC-Bokmål model. These models are fine-tuned on the Bokmål subset of NPSC plus the translated version of the Nynorsk subset, the NST, and the hesitations subset of NST. These models also replace the hesitations with single letters in the 1 billion parameters models, and the special character $\hat{\text{h}}$ shared between all types of hesitations in the 300 million parameters models since triple letters require a pad character in between.

- NPSC-Nynorsk. Since the NPSC-Nynorsk model was only available as a 300 million parameter model, this model is a 1 billion parameters version fine-tuned on the Nynorsk subset of NPSC plus the translated version of the Bokmål subset.

We trained all models for 40 epochs on a single NVIDIA RTX A6000 GPU with an effective batch size of 24 by accumulating gradients every 2 steps on a batch size of 12. The learning rate was set to $2 \cdot 10^{-5}$, with 2,000 steps of warmup and linear decaying using an Adam optimizer with $\beta_1 = 0.9$, $\beta_2 = 0.999$, and $\epsilon = 10^{-8}$. We used the PyTorch models available in the HuggingFace hub.

After fine-tuning, separate Bokmål and Nynorsk 5-gram Kneser-Ney language model were added where appropriate[5]. Two versions of the NST-NPSC model were also created, one with the Bokmål 5-gram language model, and another one with the Nynorsk language model, as we evaluate the NST-NPSC model on both subsets of NPSC. These language models were created using a combination of the training and validation sets of NPSC plus a few thousand random documents from the Norwegian Colossal Corpus (Kummervold et al., 2021, 2022). We processed a total of 78 million words by lowercasing, normalizing, and filtering out the characters that were outside the 28 Norwegian letters used for fine-tuning. We used the implementation of Kneser-Ney models (Ney et al., 1994) available in the KenLM library (Heafield, 2011). The estimation of the CTC $\alpha$ and $\beta$ values was done by grid search over {0.001, 0.01, 0.1, 0.25, 0.5, 0.75, 1, 1.5, 2, 3} on the validation set of the Bokmål subset of NPSC; we established $\alpha = 0.5$ and $\beta = 0.001$.

## 5 Results and Discussion

We evaluate the performance of the models grouping their scores by the written language of the test sets in NPSC and NST. We report word error rates as percentages[6]. For comparison purposes, we include the figures obtained in the original NPSC paper by Solberg and Ortiz (2022), as well as the work by Ortiz and Burud (2021) who also briefly evaluated ASR on NPSC. Table 3 shows the WER score of the 300 million and 1 billion parameters

---

[4]Swedish Wav2vec 2.0 large VoxRex (C) and Multilingual Wav2Vec2-XLSR-53.

[5]https://huggingface.co/NbAiLab/nb-wav2vec2-kenlm

[6]For character error rates, please see Appendix B, Tables 6, 7 and 8.

| Size | Model | NPSC | NPSC (Bokmål) | NST |
|------|-------|------|---------------|-----|
| 300M | *No language model* | | | |
| | NPSC-Bokmål | 11.76 | 9.79 | 21.46 |
| | NST | 24.50 | 22.45 | 5.52 |
| | NST-NPSC | 9.58 | 8.86 | 5.44 |
| | NST-NPSC-Bokmål | 10.37 | 8.33 | 5.49 |
| | *5-gram language model* | | | |
| | NPSC-Bokmål | 9.07 | 7.14 | 19.19 |
| | NST | 19.41 | 17.33 | **4.38** |
| | NST-NPSC | **7.60** | **6.92** | 4.39 |
| | NST-NPSC-Bokmål | 10.05 | 7.96 | 4.42 |
| 1B | *No language model* | | | |
| | NPSC-Bokmål | 9.49 | 7.51 | 17.64 |
| | NST | 25.07 | 22.94 | 5.08 |
| | NST-NPSC | 8.99 | 7.14 | 5.25 |
| | NST-NPSC-Bokmål | 8.69 | 6.46 | 4.93 |
| | *5-gram language model* | | | |
| | NPSC-Bokmål | 8.37 | 6.41 | 14.94 |
| | NST | 21.47 | 19.36 | 4.39 |
| | NST-NPSC | 8.03 | 6.15 | 4.54 |
| | NST-NPSC-Bokmål | **8.02** | **5.81** | **4.30** |
| | Ortiz and Burud (2021) | 20.64 | | |
| | Solberg and Ortiz (2022) | 17.10 | | |

Table 3: Test sets WER scores of all models fine-tuned on data containing Bokmål. Best scores in **bold** for each size.

models. In both cases, it can be seen that models trained on the Bokmål subset of NPSC perform not too well on the test set of NST. Similary, models trained only on NST underperform on the test set of the Bokmål subset of NPSC. Adding a 5-gram language model yields significant improvements across the board, ranging from a 5 points increase on the worst performing pairs of model and dataset, to a 1 point increase for the best performing pairs. However, the biggest gain in performance is the addition of extra data. The models fine-tuned on combinations of NPSC and NST produce significantly better results. On the whole NPSC, the 300M NST-NPSC model outperform Solberg and Ortiz (2022) by 9.5 points and the previous state of the art NPSC-Bokmål model by 4.16 points. For the other datasets, the 1 billion parameters model NST-NPSC-Bokmål outperformed the rest of models, yielding increases over the NPSC-Bokmål model of 0.6 points on NPSC (Bokmål) subset and of 14.89 points on NST. Interestingly, the performance of the best 300M and 1B models was very close.

An evaluation of the models for each region in

the test set of NST can also be found in Appendix C with somewhat similar results and trends. We found that there is virtually no difference in the per region performance of the models, even for the unbalanced (in terms of hours of speech in test set) regions of Oslo and Sør-Vestlandet. It is important to notice that the regions identified in NST do not reflect the diversity of spoken dialects in Norway.

For Nynorsk, as shown in Table 4, our NST-NPSC 300M model with a Nynorsk 5-gram language model attached did not beat the existing NPSC-Nynorsk 300M model. However, our newer NPSC-Nynorsk 1B model outperforms the NPSC-Nynorsk 300M model by 1.14 points.

In order to evaluate the generalization capabilities of our models, we use the Norwegian test set of FLEURS. Transcriptions on FLEURS were normalized as closely as possible to those present in the NST and NPSC, with numbers and times written out in text form. We compare the performance of our models against the Whisper models (Radford et al., 2022), which despite being architecturally different, and being trained in a supervised fashion on almost twice the amount of

| Size | Model | NPSC (Nynorsk) |
|---|---|---|
| 300M | *No language model* | |
| | NPSC-Nynorsk | 16.29 |
| | NST-NPSC | 16.52 |
| | *5-gram language model* | |
| | NPSC-Nynorsk | **12.68** |
| | NST-NPSC | 14.23 |
| 1B | *No language model* | |
| | NPSC-Nynorsk | 13.99 |
| | NST-NPSC | 26.99 |
| | *5-gram language model* | |
| | NPSC-Nynorsk | **11.54** |
| | NST-NPSC | 25.38 |

Table 4: Test sets WER scores of all models fine-tuned on data containing Nynorsk. Best scores in **bold** for each size.

hours of XLS-R and with subtitles instead of transcriptions, hold the state of the art on almost every language in FLEURS. However, it is important to notice that their WER scores are calculated on non-normalized text and their parameter counts do not match ours[7]. As shown in Table 5, our best 300 million parameters model more than doubles the performance of Whisper small (244M), with a WER of 9.88 versus 24.20. The 1 billion parameters model NST-NPSC still outperforms Whisper large by 1.53 points, and it is only a negligible 0.37 points from the version 2 of the Whisper large model, that while having 550M fewer parameters than Whisper large.

## 6 Future Work

Despite the improved performance of our models compared to the other baselines, ASR models for Norwegian still face several challenges. One major challenge is the complex phonetics and morphology of the different dialects, which makes it difficult for models to accurately transcribe the phonemes in the input speech to the correct spelling. Another challenge is the limited availability of high-quality datasets for Norwegian speech, which limits the amount of training data for ASR models.

To address these challenges, one possible to solution is to combine multiple datasets and sources of training data, such as transcribed speech and

---

[7]Whisper models are able to handle capitalization and punctuation marks.

| Size | Model | FLEURS |
|---|---|---|
| 300M | *No language model* | |
| | NPSC-Bokmål | 18.51 |
| | NST | 13.94 |
| | NST-NPSC | 12.43 |
| | NST-NPSC-Bokmål | 12.51 |
| | *5-gram language model* | |
| | NPSC-Bokmål | 12.98 |
| | NST | 11.27 |
| | NST-NPSC | 9.93 |
| | NST-NPSC-Bokmål | **9.88** |
| | Whisper small (244M) | 24.20 |
| 1B | *No language model* | |
| | NPSC-Bokmål | 16.26 |
| | NST | 13.05 |
| | NST-NPSC | 11.17 |
| | NST-NPSC-Bokmål | 11.53 |
| | *5-gram language model* | |
| | NPSC-Bokmål | 13.03 |
| | NST | 11.53 |
| | NST-NPSC | **9.87** |
| | NST-NPSC-Bokmål | 10.00 |
| | Whisper large (1.55B) | 11.4 |
| | Whisper large-v2 (1.55B) | 9.5 |

Table 5: Test sets WER scores on the Norwegian subset of FLEURS for all models. Best scores in **bold** for each size.

synthetic speech, to increase the amount of pre-training data for ASR models. With enough transcribed speech, even other more data-hungry architectures could be tested, such as Whisper.

Finally, the prospect of training wav2vec 2.0 directly on non-normalized text is an interesting avenue for research, as it would make the models directly usable without having to transform the output of the models to make them more readable.

## 7 Conclusion

In this paper, we presented several new models for automatic speech recognition of Norwegian. We evaluated these models on several datasets of Norwegian speech and compared their performance to previous work, outperforming the previous state of the art. Given that we used almost the same settings than the wav2vec 2.0 models released last year, with the addition of extra training time and data there are some interesting findings. First, adding over 400 hours of extra planned speech to the semi-improvised speech part of NPSC, performance does not plummet, but actually increases

from 6.41 to 5.81 WER for Bokmål in the 1B settings. The 300M model seems more sensitive in this regard and the WER decreases from 7.14 to 7.96 WER. For NST, the trend is exactly the same, although the differences are smaller.

Interestingly, the out of domain performance of the models is also greatly improved by adding the planned speech in NST to NPSC. Models on both sizes increase their WER scores from 12.98 to 9.88 for the 300M model, and from 13.03 to 9.87 for the 1B model.

We are releasing our best performing models and evaluation code for replicability, and hope to contribute to the advance of ASR for Norwegian.

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

## A  Availability

The best performing models and the code use to train and evaluate them are released with a permissive license in a model hub:

- NST-NPSC 300M model as
  `nb-wav2vec2-300m-bokmaal-v2`.

- NST-NPSC-Bokmål 1B model as
  `nb-wav2vec2-1b-bokmaal-v2`.

- NPSC-Nynorsk 300M model as
  `nb-wav2vec2-300m-nynorsk`.

- NPSC-Nynorsk 1B model as
  `nb-wav2vec2-1b-nynorsk`.

The results raw data is also available in a code repository to replicate all tables and figures in this work at `nb-wav2vec2`.

## B  Character Error Rates (CER)

| Size | Model | NPSC (Nynorsk) |
|------|-------|----------------|
| 300M | *No language model* | |
| | NPSC-Nynorsk | 4.91 |
| | NST-NPSC | 5.03 |
| | *5-gram language model* | |
| | NPSC-Nynorsk | **4.38** |
| | NST-NPSC | 4.80 |
| 1B | *No language model* | |
| | NPSC-Nynorsk | 4.52 |
| | NST-NPSC | 7.33 |
| | *5-gram language model* | |
| | NPSC-Nynorsk | **4.12** |
| | NST-NPSC | 7.07 |

Table 6: Test sets CER scores of all models fine-tuned on data containing Nynorsk. Best scores in **bold** for each size.

| Size | Model | FLEURS |
|------|-------|--------|
| 300M | *No language model* | |
| | NPSC-Bokmål | 4.96 |
| | NST | 3.96 |
| | NST-NPSC | 3.48 |
| | NST-NPSC-Bokmål | 3.46 |
| | *5-gram language model* | |
| | NPSC-Bokmål | 3.83 |
| | NST | 3.46 |
| | NST-NPSC | 2.92 |
| | NST-NPSC-Bokmål | **2.89** |
| 1B | *No language model* | |
| | NPSC-Bokmål | 4.42 |
| | NST | 3.88 |
| | NST-NPSC | 3.13 |
| | NST-NPSC-Bokmål | 3.24 |
| | *5-gram language model* | |
| | NPSC-Bokmål | 3.73 |
| | NST | 3.58 |
| | NST-NPSC | **2.89** |
| | NST-NPSC-Bokmål | 2.91 |

Table 7: Test sets CER scores on the Norwegian subset of FLEURS for all models. Best scores in **bold** for each size.

| Size | Model | NPSC | NPSC (Bokmål) | NST |
|------|-------|------|---------------|-----|
| | *No language model* | | | |
| | NPSC-Bokmål | 3.63 | 3.13 | 5.05 |
| | NST | 8.84 | 8.23 | 1.75 |
| | NST-NPSC | 3.08 | 2.87 | 1.70 |
| | NST-NPSC-Bokmål | 3.07 | 2.55 | 1.76 |
| 300M | *5-gram language model* | | | |
| | NPSC-Bokmål | 3.24 | 2.74 | 4.59 |
| | NST | 8.17 | 7.53 | **1.55** |
| | NST-NPSC | **2.83** | **2.62** | 1.52 |
| | NST-NPSC-Bokmål | 3.31 | 2.75 | 1.56 |
| | *No language model* | | | |
| | NPSC-Bokmål | 3.17 | 2.67 | 4.23 |
| | NST | 9.32 | 8.65 | 1.63 |
| | NST-NPSC | 2.99 | 2.54 | 1.65 |
| | NST-NPSC-Bokmål | 2.71 | 2.06 | 1.64 |
| 1B | *5-gram language model* | | | |
| | NPSC-Bokmål | 3.01 | 2.51 | 3.69 |
| | NST | 8.75 | 8.09 | 1.52 |
| | NST-NPSC | 2.85 | 2.39 | 1.53 |
| | NST-NPSC-Bokmål | **2.62** | **1.98** | **1.53** |

Table 8: Test sets CER scores of all models fine-tuned on data containing Bokmål. Best scores in **bold** for each size.

## C   NST regions

| Region | Train | | Test | |
|--------|-------|---------|-------|---------|
| | **Hours** | **Samples** | **Hours** | **Samples** |
| Oslo-området | 53.2 | 38,688 | 25.3 | 17,729 |
| Ytre Oslofjord | 48.0 | 34,008 | 7.3 | 4,935 |
| Bergen og Ytre Vestland | 45.7 | 31,824 | 8.3 | 5,922 |
| Sør-Vestlandet | 42.2 | 29,328 | 10.3 | 6,909 |
| Trøndelag | 38.4 | 27,456 | 9.3 | 5,922 |
| Sørlandet | 36.9 | 26,600 | 9.0 | 5,922 |
| Voss og omland | 33.6 | 22,776 | 9.4 | 5,922 |
| Troms | 30.5 | 19,344 | 9.6 | 4,935 |
| Nordland | 28.0 | 20,591 | 8.8 | 5,922 |
| **Total** | **411.5** | **289,934** | **115.3** | **75,965** |

Table 9: Distribution of number of hours and speakers for each of the dialect regions (region of youth) of the Norwegian subset of the NST dataset.

| Size | Region | NPSC-Bokmål | NPSC-Nynorsk | NST | NST-NPSC | NST-NPSC-Bokmål |
|---|---|---|---|---|---|---|
| | *No language model* | | | | | |
| | Bergen og Ytre Vestland | 26.14 / 6.24 | 45.57 / 11.34 | 6.18 / 1.77 | 5.92 / 1.75 | 5.92 / 1.75 |
| | Hedmark og Oppland | 19.22 / 4.08 | 42.36 / 10.31 | 4.71 / 1.12 | 4.56 / 1.06 | 4.56 / 1.14 |
| | Nordland | 21.92 / 4.67 | 42.62 / 10.03 | 5.21 / 1.27 | 5.00 / 1.20 | 4.99 / 1.25 |
| | Oslo-området | 20.21 / 5.90 | 42.50 / 11.87 | 6.67 / 3.29 | 6.60 / 3.21 | 6.65 / 3.26 |
| | Sunnmøre | 22.72 / 5.00 | 41.56 / 9.64 | 5.02 / 1.16 | 5.04 / 1.15 | 5.10 / 1.21 |
| | Sør-Vestlandet | 24.55 / 5.78 | 45.44 / 11.53 | 6.23 / 1.57 | 6.13 / 1.53 | 6.25 / 1.60 |
| | Sørlandet | 21.99 / 4.77 | 44.04 / 10.52 | 5.52 / 1.34 | 5.45 / 1.31 | 5.48 / 1.37 |
| | Troms | 21.66 / 4.35 | 42.56 / 9.71 | 4.21 / 0.97 | 4.25 / 0.95 | 4.28 / 1.01 |
| | Trøndelag | 18.28 / 3.77 | 40.26 / 9.54 | 4.32 / 1.02 | 4.27 / 1.00 | 4.43 / 1.07 |
| | Voss og omland | 20.22 / 4.32 | 38.84 / 8.86 | 4.10 / 0.98 | 4.21 / 0.98 | 4.24 / 1.04 |
| | Ytre Oslofjord | 21.08 / 4.69 | 44.45 / 11.36 | 6.04 / 1.54 | 5.87 / 1.46 | 5.94 / 1.50 |
| 300M | *5-gram language model* | | | | | |
| | Bergen og Ytre Vestland | 22.75 / 5.58 | 42.64 / 10.81 | 4.91 / 1.52 | 4.83 / 1.54 | 4.70 / 1.55 |
| | Hedmark og Oppland | 17.69 / 3.75 | 39.31 / 9.81 | 3.58 / 0.94 | 3.48 / 0.88 | 3.58 / 0.96 |
| | Nordland | 19.59 / 4.20 | 39.76 / 9.58 | 3.89 / 1.04 | 3.89 / 1.00 | 3.91 / 1.04 |
| | Oslo-området | 18.39 / 5.53 | 39.48 / 11.36 | 5.70 / 3.10 | 5.66 / 3.04 | 5.67 / 3.07 |
| | Sunnmøre | 19.74 / 4.39 | 38.94 / 9.25 | 3.90 / 0.98 | 4.04 / 0.99 | 4.03 / 1.04 |
| | Sør-Vestlandet | 21.44 / 5.17 | 42.45 / 10.97 | 4.74 / 1.31 | 4.84 / 1.31 | 4.87 / 1.36 |
| | Sørlandet | 19.64 / 4.30 | 41.34 / 10.03 | 4.24 / 1.12 | 4.29 / 1.10 | 4.29 / 1.14 |
| | Troms | 19.10 / 3.87 | 39.90 / 9.31 | 3.17 / 0.79 | 3.26 / 0.78 | 3.35 / 0.85 |
| | Trøndelag | 16.78 / 3.48 | 36.68 / 8.94 | 3.34 / 0.85 | 3.38 / 0.84 | 3.55 / 0.92 |
| | Voss og omland | 18.56 / 3.97 | 36.18 / 8.50 | 3.32 / 0.85 | 3.36 / 0.83 | 3.37 / 0.89 |
| | Ytre Oslofjord | 18.53 / 4.14 | 40.97 / 10.75 | 4.51 / 1.26 | 4.53 / 1.21 | 4.57 / 1.24 |
| | *No language model* | | | | | |
| | Bergen og Ytre Vestland | 21.88 / 5.41 | 43.60 / 11.38 | 5.47 / 1.61 | 5.85 / 1.73 | 5.17 / 1.61 |
| | Hedmark og Oppland | 14.48 / 3.06 | 39.23 / 9.89 | 4.29 / 1.02 | 4.29 / 0.99 | 4.15 / 1.04 |
| | Nordland | 17.78 / 3.79 | 40.67 / 9.97 | 4.44 / 1.08 | 4.68 / 1.11 | 4.39 / 1.10 |
| | Oslo-området | 16.70 / 5.10 | 40.45 / 11.58 | 6.27 / 3.15 | 6.30 / 3.12 | 6.14 / 3.16 |
| | Sunnmøre | 20.41 / 4.57 | 39.73 / 9.67 | 4.59 / 1.07 | 5.19 / 1.19 | 4.51 / 1.09 |
| | Sør-Vestlandet | 20.84 / 5.02 | 43.82 / 11.54 | 6.23 / 1.55 | 6.19 / 1.55 | 6.03 / 1.54 |
| | Sørlandet | 17.69 / 3.72 | 41.63 / 10.26 | 5.23 / 1.29 | 5.30 / 1.27 | 4.92 / 1.24 |
| | Troms | 17.22 / 3.42 | 40.30 / 9.44 | 3.75 / 0.86 | 3.92 / 0.87 | 3.47 / 0.83 |
| | Trøndelag | 14.16 / 3.01 | 37.28 / 9.23 | 3.80 / 0.89 | 4.07 / 0.91 | 3.65 / 0.90 |
| | Voss og omland | 16.50 / 3.52 | 36.05 / 8.62 | 3.72 / 0.90 | 4.04 / 0.93 | 3.78 / 0.93 |
| | Ytre Oslofjord | 17.69 / 3.83 | 43.30 / 11.26 | 5.32 / 1.38 | 5.47 / 1.36 | 5.25 / 1.39 |
| 1B | *5-gram language model* | | | | | |
| | Bergen og Ytre Vestland | 18.34 / 4.67 | 41.20 / 10.92 | 4.69 / 1.49 | 5.03 / 1.58 | 4.54 / 1.50 |
| | Hedmark og Oppland | 12.37 / 2.64 | 36.81 / 9.49 | 3.65 / 0.93 | 3.61 / 0.88 | 3.63 / 0.96 |
| | Nordland | 14.97 / 3.24 | 38.22 / 9.55 | 3.68 / 0.96 | 3.88 / 0.97 | 3.70 / 0.99 |
| | Oslo-området | 14.53 / 4.68 | 37.83 / 11.11 | 5.67 / 3.04 | 5.71 / 3.01 | 5.57 / 3.05 |
| | Sunnmøre | 16.79 / 3.82 | 37.81 / 9.36 | 3.91 / 0.96 | 4.46 / 1.07 | 3.88 / 1.00 |
| | Sør-Vestlandet | 17.63 / 4.32 | 41.68 / 11.09 | 5.30 / 1.41 | 5.31 / 1.40 | 5.19 / 1.41 |
| | Sørlandet | 14.95 / 3.19 | 39.27 / 9.86 | 4.39 / 1.16 | 4.46 / 1.13 | 4.16 / 1.11 |
| | Troms | 14.54 / 2.95 | 37.90 / 9.05 | 3.16 / 0.76 | 3.27 / 0.77 | 3.04 / 0.77 |
| | Trøndelag | 11.86 / 2.54 | 34.62 / 8.70 | 3.27 / 0.80 | 3.44 / 0.80 | 3.11 / 0.81 |
| | Voss og omland | 13.43 / 2.93 | 34.14 / 8.36 | 3.19 / 0.83 | 3.51 / 0.84 | 3.23 / 0.85 |
| | Ytre Oslofjord | 15.36 / 3.35 | 40.32 / 10.72 | 4.42 / 1.22 | 4.64 / 1.20 | 4.41 / 1.24 |

Table 10: Per region test set word and character error rates (WER / CER) of all models fine-tuned on NST.