# OpenReview forum: "Boosting Norwegian Automatic Speech Recognition"
_NoDaLiDa/2023/Conference — NoDaLiDa 2023_

### Official Review · Reviewer_fsFx · 2023-03-09
**Boosting Norwegian Automatic Speech Recognition**

**Rating:** 8
**Confidence:** 5

**Review:**

A well written paper presenting results from fine-tuning large wav2vec2 models on different available Norwegian datasets. The presented results show very high-performing systems for tasks ranging from read speech (NST and FLEURS data) to more the more mixed style speech from the Q&A sessions of the Norwegian Parliament (NPSC corpus), where parts of the spoken content is based on manuscripts, while other parts can be ad-libbed.

The main novelty of the paper lies in the reported performance, and the use of very large models (300 - 1000 million parameters), and a wide-span 5-gram language model.

N-gram language models for speech recognition are normally word-based, i.e. they provide the conditional probability of a vocabulary word given the $N-1$ preceding words. The paper states that the 5-gram model employed was trained on a pre-processed text corpus consisting of 78 million words. That seems extremely small for training a "normal" 5-gram model, even with Kneser-Ney smoothing (a 38 word vocabulary will produce 79 million unique 5-word combinations). So, is the 5-gram model a character level language model or a sub-word token model rather than a word-based language model?

Some more detailed comments:
>The Nordic Språkteknologi (NST) dataset is a multi-lingual speech recognition dataset that contains a total 540 hours of speech in Swedish, Danish and Norwegian Bokmål

The NST data is really three separate datasets, one per language, that are structurally very similar. The 540 hour quantity is approximately correct **per language** (cfr. e.g. the specification of the Norwegian dataset further down in the same paragraph).

At the end of Section 4, it is referred to "the 28 Norwegian letters" - Norwegian has 29 letters in the alphabet

Regarding the comparison with Whisper performance on the same data. There are three main differences to be noted,
1. the architectural differences
2. the number of parameters
3. the amount of fine-tuning
Whisper is trained on multilingual data, including Norwegian (over 200hrs) in a semi-supervised setting. The wav2vec architectures are trained on multilingual or Swedish data, self-supervised, with fine-tuning on Norwegian data both in-domain and out-of-domain. It is likely that a significant part of the difference in performance is due to the fine-tuning to the Norwegian language.  Moreover, the issue of a trained language model is important.

Some of the references need editing, e.g.
>Jean-Pierre Martens and Belgium Fpm. 2000. Final  report of cost action 249 continuous speech recognition over the telephone country acronym name of institution.

Prof. Martens is the sole author (editor), the latter part (ountry acronym name of institution) can be omitted.

Also, reference to the reorganised NST database should be to Språkbanken, not to NST, and e.g. include an url, similar for NPSC.  The Whisper paper lacks publication info.

**Paper Type:**

Long paper

---

### Official Review · Reviewer_kczL · 2023-03-10
**Significant performance improvement, but no insights into why NST and NPSC are complimentary across test sets**

**Rating:** 7
**Confidence:** 4

**Review:**

This paper presents work done to combine NST and NPSC data sets to fine-tune XLS-R and VoxRex ASR models and the authors present good results and strong improvements.

The grammar of the paper is good, I only noticed 1-2 spelling errors or typos, but clarity would improve with more sections, especially a Data section and an Experimental setup section. Specifically, there is no explicit mention of the software used for training and testing, which is relevant since this is primarily a engineering paper. You correctly cite the KenLM toolkit, but not the ASR toolkit(s).

Scientifically, there is little interest in the finding that 'performance increases with more data'. What would be interesting is an error analysis of the errors made by the different ASR systems and a comparison to get to an answer to why these two datasets are complimentary.

From a methodological POV, I think it is important to have the WER performance of Vanilla VoxRex/XLS-R to compare against in the tables of results and for results that are very close to calculate statistical significance. I am also curious why Whisper performance on NST and NPSC test sets is not reported?  Also, you should make explicit that best performance (bold-faced type in tables) are based on WER performance and not CER.

In summary, the performance improvements you obtain are considerable and impressive, but if you intend to write an engineering type paper, we need more implementation details in the paper. If you want to focus more on the scientific part then insights to when and why read-aloud/planned speech can compliment ASR performance on more challenging speech genres would be _really_ interesting!

**Paper Type:**

Long paper

---

### Official Review · Reviewer_dsLU · 2023-03-10
**The paper has impressive results, but a messy presentation of them**

**Rating:** 7
**Confidence:** 3

**Review:**

This paper presents very good results for Norwegian ASR tasks. It is positive that Whisper is used as a baseline. The work appears to be mainly a straightforward application of the wav2vec 2.0 architecture to Norwegian datasets, but although it does not represent any methodical innovation, I believe it would still be a valuable addition to the research literature. It is my recommendation that it be accepted for publication after editing to remedy some of the following problems.

It is unclear why the title is "_Boosting_ Norwegian ASR". Boosting is most commonly associated with a specific machine learning method, which is not employed here. I thus assume "boosting" is used in the sense of "helping something to improve", but perhaps this should be expanded upon in the introduction and/or conclusion – or the title be changed.

The description of the ASR model used should be more thorough, and preferably include a figure.

The presentation of the results (in 8 separate tables, in addition to 3 more in the appendix) seems quite messy. It currently gives a particularly bad impression that the first two tables referenced in the results section are in the appendix. (Although I did not realize this was the appendix at first, as the heading was left back on page 8. It should be moved to stand above the first table in the appendix.) I now realize that this reference to Tables 10 and 11 on line 405, is probably a typo, and meant to be Tables 3 and 4. This mistake highlights the fact that there should probably be fewer tables. E.g., Tables 7 and 8 seem to only be referenced together – shouldn't they perhaps be merged to one larger table?

I also think the table captions should specify that the word/character error rates are in percentages, and it is unfortunate that the abbreviation CER is not defined anywhere.

Other comments:

- 011: NPSC and WER should be written out

- 100: Wrong use of quotes (only close quotes are used, no open quotes). It might also be good to include the Norwegian for these two terms.

- 123: When were the newer datasets introduced?

- 131: «defunct» rather than «extinct»?

- 127–152: Shouldn’t there be a reference here? No year is given.

- 168: already split by who?

- 237: wav2vec 2.0 improves

- 295: Not obvious what you mean by «topological closeness».

- 370: Write $2 \cdot 10^{-5}$ rather than «2e-5». Applies throughout.

- 371: Missing spaces around =. Applies throughout.

- 383: the a

- 538: despite being

- 552: "supervisely" is not a word. "Supervisedly" is used, but it is uncommon.

- 556--558: This is unclear.

- 564: ~~it's~~ it is

- 565: rephrase "from its v2"

- 649: out-of-domain performance

- 652: decrease [or improve] their WER scores

References:

- Several instances of the same authors being referenced in different ways: Ingunn Amdal (662, 665, 731), Per E. Kummervold (706 and 713).

- Missing periods after initials throughout.

- Wrong case many places: Xls-r, Kenlm, Fpm, Bert, htk, asr, norwegian, ...

- 748: URL?

- 776: double comma

**Paper Type:**

Long paper

---

### Decision · Program_Chairs · 2023-03-17

Accept